# Me-ifestos for Visualization Empowerment in Teaching (and Learning?)

A submission to alt.VIS 2022

With contributions from participants at Dagstuhl Seminar 22261 (June 2022)

## Abstract

One sunny June afternoon in a remote castle deep in Saarland, a group of visualization researchers of many stripes — from different countries, disciplines and generations — came together to discuss  teaching and learning for empowerment in visualization. Our first order of business in Saarland was to develop a shared vision: a manifesto of sorts that would guide us towards strategies to broaden data visualization skills, make them more common and accessible, and enable this empowerment. That exercise failed. Instead of creating one common manifesto (who are we to do that, anyway?), we took a different, more personal approach. We found things we liked and things we didn't: inspiration and counter-inspiration, we crafted individual value and commitment statements ("me-ifestos"), we shared these with each other and reflected on them. The process of writing these commitment statements was illuminating and motivating. It was a positive experience with such visceral effects that we want to share the feeling with the visualization community. And so we offer a collection of "me-ifesto" excerpts from the authors and a call to action: we invite you to craft and divulge your own me-ifesto.

# Me-ifestos for Visualization Empowerment in Teaching (and Learning?)

Contributors:
(from Dagstuhl Seminar 22261)

Jan Aerts
Amador Bioscience – Hasselt

Wolfgang Aigner
St. Pölten University of Applied Sciences

Benjamin Bach
University of Edinburgh

Fearn Bishop
BBC

Magdalena Boucher
St. Pölten University of Applied Sciences

Peter C.-H. Cheng
University of Sussex

Alexandra Diehl
Universität Zürich, CH

Jason Dykes
City – University of London

Sarah Hayes
Munster Technological University

Uta Hinrichs
University of Edinburgh

Samuel Huron
Institut Polytechnique de Paris

Christoph Kinkeldey
Hamburg University of Applied Sciences

Andy Kirk
Visualising Data

Søren Knudsen
IT University of Copenhagen

Doris Kosminsky
Federal University of Rio de Janeiro

Tatiana Losev
Simon Fraser University

Areti Manataki
University of St Andrews

Andrew Manches
University of Edinburgh

Isabel Meirelles
OCAD University

Luiz Morais
INRIA

Till Nagel
Hochschule Mannheim

Rebecca Noonan
Munster Technological University

Georgia Panagiotidou
University College London

Laura Pelchmann
Universität Köln

Fateme Rajabiyazdi
Carleton University

Christina Stoiber
St. Pölten University of Applied Sciences

Tatiana von Landesberger
Universität Köln

Jagoda Walny
Canada Energy Regulator

Wesley Willett
University of Calgary

One sunny June afternoon in a remote castle deep in Saarland, a group of visualization researchers of many stripes — from different countries, disciplines and generations — came together to discuss teaching and learning in visualization. We had built democratising websites to make datasets openly available [12, 22], to encourage discourse in communities, posted pre-prints online so that people could read our work for free and felt the warm glow of satisfaction that comes from a student getting the data vis bug. We thus found our common desire to empower the people with whom we interact so that they can understand and use data in their lives. Our first order of business in Saarland was to develop a shared vision: a manifesto of sorts that would guide us towards strategies to broaden data visualization skills, make them more common and accessible, and enable this empowerment.

That exercise failed.

It did not fail because we could not agree. It failed because what we felt most strongly about reflected our individual contexts, values, and goals. Some of us were university lecturers leading modules with varied numbers of students at different levels, some were teaching assistants with less agency and more focused responsibilities, others taught informally through research work or in large organizations. Some of us came at this from a perspective of enabling democracy, some with a passion for the visual and a desire to use this as inspiration for others, some with a broad goal of empowerment and understanding for all.

As is often the case, out of our initial failure emerged something that we found thought-provoking and worth sharing with the VIS community. Instead of creating one common manifesto (who are we to do that, anyway?), we took a different, more personal approach. Instead of adding another manifesto to the collection of excellent and inspiring manifestos that exist, we took time to read them, process them, discuss them and anchor some individual and collective reflec-

tion. We thought about their contents, compared their approaches, decided what they taught us about ourselves, our own experiences and attitudes, and identified statements that resonated with us individually. We used this thinking to draft individual visualization "me-ifestos", or *personal commitments about how to empower others through data visualization*. We took some time, thought some more, and then shared these drafts with one another.

What surprised us was how deeply revealing the ensuing discussion was about our values and desires for the world. Reflecting on what it means to empower people made us ask, *what do we want to empower people to do?*, *how can we best do that?*, and even *what does best mean anyway?* The answers highlighted the situated understandings of *empowerment*, ranging from constructivist teaching and physicalization, to co-design and policy intervention. The variety in our approaches reflected the variety in our backgrounds, and the different situations through which we personally felt we could approach the task of strengthening visualization empowerment in others.

Seeing others' statements helped us understand one another and ourselves in ways that surprised us. It made us want to revise some of our own statements, or add new ones. Instead of creating one common manifesto, we created a shared collection of value statements and commitments that we could mix and remix to help ourselves gain clarity about why we cared about empowering people with data visualization — and what we might prioritize to do so: a collective manifesto of commitments.

We found this activity to be incredibly freeing, reflexive, and thought provoking. It was a learning excercise for and about ourselves as teachers. Sharing the statements also brought us together from across our various practices. This was a positive experience with such visceral effects that we want to discuss it with the visualization community. And so in the following pages, we share a

reflection on manifestos, a collection of "me-ifesto" excerpts from the authors and importantly, **a call to action**: a request that you craft and share your own.

## 1  MANIFESTOS IN DESIGN AND VIS

Manifestos are used widely in art and design to express approaches and attitudes, to develop genres through design constraint, to develop institutional, disciplinary, and even individual cultures and to direct as educational devices that may guide learning. Tom Nelson's "DesignManifestos.org" is a rich and beautifully curated site that provides access to a wide selection of these [17].

Art and design education would seem to have plenty in common with visualisation education and so some of this activity is likely relevant to our domain. For example, Ainslie Hunter's "Courses That Matter" manifesto (Figure 1a) offers a perspective on learning that reminds us of the underlying value of what we do as educators.

Figure 1: Excerpts from: (a) Ainslie Hunter's *Courses That Matter* manifesto [10]; (b) Sister Corita Kent's *Immaculate Heart College Art Department Rules* [11]; (c) Marion Deuchars' contribution to *The New Art School Rules!* exhibition [20]

They are generative too. Sister Corita Kent's celebrated rules inspired Design Manchester to curate "The New Art School Rules!" [20] (see Figure 1b), a "collective manifesto" consisting of statements submitted by contemporary artists and designers.

### No More Manifestos

But is there a need for manifestos in the first place? In an accessible and thought-provoking whirlwind tour of the *Manifesto!*, BBC Radio 4's *No More Manifestos* [1] offers a series of perspectives that frame manifestos as prescriptive, demanding, "*aggressive impositions of ideas*" (Gilbert & George [1]) that "*sound bombastic, they sound anachronistic*" (Jennifer Higgie [1]). Indeed, we are warned that manifestos can be considered to be "*deeply problematic [in the 21st century and] could be moving towards what we could call fascism*" (Tom McCarthy [1]). Perhaps *No More Manifestos* is an important call to inaction.

### Many More Manifestos

But in their denouncement of the political manifesto these contributors to *Manifesto!* offer an opportunity. Gilbert & George ressure us in reminding us that their *X Commandments* are *FOR* Gilbert and George: "*We never want to tell other people how to behave or what to do, our vision is for ourself*" [7]. McCarthy qualifies his concerns as relating to "*non-ironic*" manifestos. He reminds us that irony is subversive and that ironic manifestos through their radical authenticity allow us to create cultural disturbances and are "*a proper response to our age*".

And here we begin to see that the *intent*, and the *recipients* of the ideas and the extent to which they are *imposed* are important factors

in differentiating between manifestos that are mandatory and used to exert power that affects others, and those that are optional, voluntary and might be used to empower the individual.

Gilbert & George are among many designers, artists and educators who develop manifestos that describe their values, motives and intent [17]. Sometimes these declarations are calls to action, efforts to inspire [11]. Sometimes they constrain creative options to develop styles and genres [24]. Frequently they are blatantly ironic, leveraging the clarity and authority of the political manifesto to present one's artistic identity in a manner that simultaneously paints a sharp picture of a position, smacks of pomposity and celebrates the self-mocking, while perhaps (indeed hopefully) undermining the sincerity of the original form. Indeed, in *No More Manifestos* Grayson Perry can't resist a snigger when suggesting with the ultimate irony that "*Every Artist Should Have Their Own Manifesto*" [1].

These perspectives take us somewhere. To the *personal manifesto*. The explicit statement of one's own values, intentions and motives. Reassuringly it kills any efforts (explicit at least) to impose. It may be empowering. It may be motivating. It may be an effective communication device that might help individuals collaborate. It may be what we found in Saarland. The personal manifesto makes no demands of others, but may be empowering to the author and may be inspiring to those who may adopt it, or reject it, or use parts of it, or question it, or decide to spend time developing a similarly clear declaration of their own.

Evan Peck has one [18]. In *A Student-Centered Research Manifesto* he outlines guiding principles for an inclusive research environment. By developing these in discussion and with colleagues and existing statements he presents a sharp focus on his intentions, and his efforts and failures to meet the high standards that he has set. It is personal, public, testing, and likely empowering. It is certainly deeply inspiring. This inspiration, and some discussion of the politics, power, past and potential of personal manifestos initiated some creative thinking and actions on personal manifestos in Saarland this summer, as we considered their potential for empowering teachers and learners in visualization education:

- Maybe "*Every **Teacher** Should Have Their Own Manifesto*"?

- Maybe "*Every **Designer** Should Have Their Own Manifesto*"?

- Maybe "*Every **Data Vizzer** Should Have Their Own Manifesto*"?

### Me-ifestos for Self-Reflection

It certainly felt interesting to find out whether we could draft manifestos, what they might look like, how the process might work and whether the documents, the activity and the thinking that took place had personal or pedagogic benefit.

Without necessarily knowing all of the above during the Dagstuhl workshop, we first simply gave ourselves time to think and sketch out our own ideas for manifestos. Upon coming together to discuss, we acknowledged that manifestos are a group's proclamations of virtues, values, and actions, often imposing, often exclusionary, most often aiming to disrupt the status quo and for others to follow the group manifesto. We found ourselves agreeing on a way to subvert manifesto writing from its public followship to a deeply personal and individual activity. We thus shifted to creating *me-ifestos*.

As we progressed, it became clear that we were speaking about both values and aspirations, and feeling inspired by sharing both. Eventually it seemed to make it easier to divide the process into creating belief statements ('I believe..') which are easy, followed by commitments to action ('I will..') - which are of course much more difficult. Such a separation might seem counterintuitive for manifestos, which are meant to signify both intent and action. However, accepting the gap between intent and action allowed a certain self-compassion to emerge in this exercise, permitting ourselves to occasionally fall short in some of our actions.

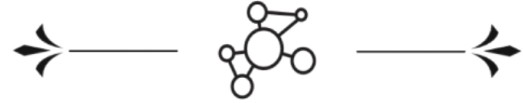

# Me-ifesto Exercise

**COFFEE**: Stop — get a drink, find an hour to commit to some thinking.

**CONSIDER**: What do you value? What do you believe in? Read your own manifesto if you have one.

**CHECK**: Read through some of the manifestos we have found and personal statements we have crafted. Find others for yourself. Which of these statements resonate and which don't?

**CRAFT**: Adopt, adapt, or create some statements that describe your values

**COMMIT**: Make personal commitments to actions or intentions that guide you to empower people through visualization.

Feel free to use, borrow, or remix existing manifestos and add your own.

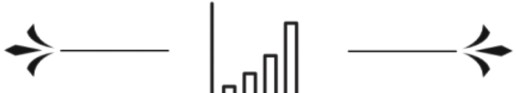

---

### ME-IFESTO MANUFACTURE

Making a me-ifesto? It's straightforward, so let's get down to it!

We'd like you to find an hour in your day to engage with the *Me-ifesto exercise* that you can see right here (left, here) by stopping, and considering what values and/or commitments you want to make to your visualization practice.

The more voices are heard, and the more perspectives seen, the more powerful this exercise becomes. *So we really need you!*

Where do you start from?
Well, maybe you have explicit statements that describe personal commitments to actions or intentions that you use to guide your efforts to empower people through visualization?
If so, then take these statements, reflect on them and revise them through our proposed process.
If not, well, that's interesting!
Why not?
We invite and indeed encourage you to spend an hour producing Draft 0, which you can revise as you develop and as your experiences and perspectives and statements shape your approach.

And you know what?
We want you to do this **RIGHT NOW**.
Really!
Right away!
It will take you an hour. So use the next hour!
The one you were going to spend reading this paper (which turns out to be a bit shorter than you were expecting).
At the end of this paragraph we want you to stop reading.
And then follow the big yellow arrow and start doing the *Me-ifesto exercise* by following the 5 steps.
Firstly, **STOP READING** and *get coffee* (or something). You likely have a nice coffee machine. Or if not, there's that place down the road — the one with the friendly barista and the tempting overpriced cakes. Head there now and get a drink! (and a cake – treat yourself)
Because, we're going to get you to do something you don't usually do when reviewing or reading a paper once you have calmed down and made some thinking space and have your drink in hand.
We're gonna get you to do some *THINKING*!
About yourself, and your role, and your values and how you fit in.
And then we're gonna get you to do some *WRITING*!
About your values, intentions, hopes and expectations, to help you develop your own statements and commitments.

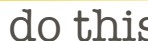

do this

# Inspiration and Counter-Inspiration

We collectively drew upon many other manifestos to help us articulate our values and commitments. We found things we liked and things we didn't: inspiration and counter-inspiration. We do not condone all of these.

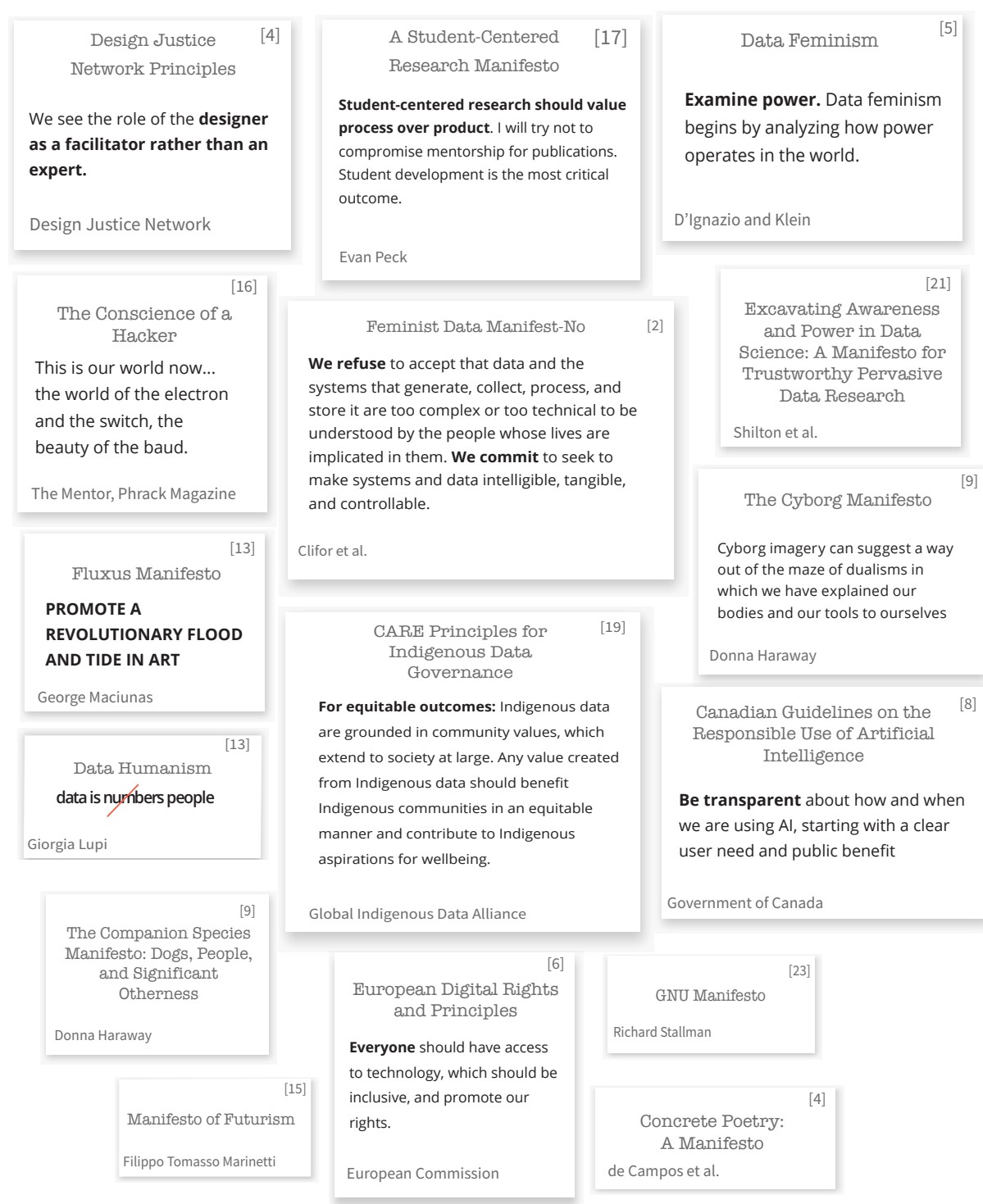

## Design Justice Network Principles [4]

We see the role of the **designer as a facilitator rather than an expert.**

Design Justice Network

## A Student-Centered Research Manifesto [17]

**Student-centered research should value process over product**. I will try not to compromise mentorship for publications. Student development is the most critical outcome.

Evan Peck

## Data Feminism [5]

**Examine power.** Data feminism begins by analyzing how power operates in the world.

D'Ignazio and Klein

## The Conscience of a Hacker [16]

This is our world now... the world of the electron and the switch, the beauty of the baud.

The Mentor, Phrack Magazine

## Feminist Data Manifest-No [2]

**We refuse** to accept that data and the systems that generate, collect, process, and store it are too complex or too technical to be understood by the people whose lives are implicated in them. **We commit** to seek to make systems and data intelligible, tangible, and controllable.

Clifor et al.

## Excavating Awareness and Power in Data Science: A Manifesto for Trustworthy Pervasive Data Research [21]

Shilton et al.

## The Cyborg Manifesto [9]

Cyborg imagery can suggest a way out of the maze of dualisms in which we have explained our bodies and our tools to ourselves

Donna Haraway

## Fluxus Manifesto [13]

**PROMOTE A REVOLUTIONARY FLOOD AND TIDE IN ART**

George Maciunas

## CARE Principles for Indigenous Data Governance [19]

**For equitable outcomes:** Indigenous data are grounded in community values, which extend to society at large. Any value created from Indigenous data should benefit Indigenous communities in an equitable manner and contribute to Indigenous aspirations for wellbeing.

Global Indigenous Data Alliance

## Canadian Guidelines on the Responsible Use of Artificial Intelligence [8]

**Be transparent** about how and when we are using AI, starting with a clear user need and public benefit

Government of Canada

## Data Humanism [13]

data is numbers people

Giorgia Lupi

## The Companion Species Manifesto: Dogs, People, and Significant Otherness [9]

Donna Haraway

## European Digital Rights and Principles [6]

**Everyone** should have access to technology, which should be inclusive, and promote our rights.

European Commission

## GNU Manifesto [23]

Richard Stallman

## Manifesto of Futurism [15]

Filippo Tomasso Marinetti

## Concrete Poetry: A Manifesto [4]

de Campos et al.

I believe that visualization is both a process and an outcome and that both are valuable conclusions to projects.

I believe that everyone has a right to see data in a way that resonates with them, to see and reason about the world in new ways.

I want to support learners in understanding the process of designing data visualizations and get to know different tools they can use out of the university context. It's **not about the results**. It's about the process to the final results and what they learned on the way towards it.

I want to enable learners to think out of the box and realize how they can leverage previous personal interests and talents in the context of new topics they are learning to enhance engagement, spark curiosity and give them the chance to **expand their horizon** beyond what can be taught in time-limited courses.

I will endeavour to expose my biases, knowledge gaps and assumptions.

I recognize that data, software, models, and their outputs (including visualizations) are **never entirely neutral** or objective.

I will advocate for designers, instructors and learners to probe the **deeper conceptual structure** of their target to-be visualised topic beyond the overt structure of the given data.

I will **reach out** to a citizen science project to find out what they need from visualization.

Eu me comprometo a promover um pensamento crítico e a considerar diferentes corpos no ensino, aprendizagem e criação de visualizações de dados.

I will endeavour to demonstrate (through my practice) and espouse (through my teaching) the obligation to be **truthful, trustworthy, and accessible**, in all data visualisation design.

I will **revisit and develop** my commitments and guiding statements in light of their use and those of others.

# I am committed to continuously reflect on my own role in data visualization including its ethical and societal aspects and inspire others to do so.

As an educator and a designer, I commit to spreading the idea that data visualization is created by people, for people, and about people, but can — and should — include non-human beings. I also acknowledge that people have strengths, limits, and biases.

I will use data visualization as a **process** to develop meaningful questions in both teaching and learning.

# I endeavour to encourage children's experiences of data visualization by balancing creativity, play and education.

I endeavour to help people see their data and their views in a broader context.

On a foundational level, **I believe in the student**'s eagerness to learn, their passion to explore, and their capabilities to create.

J'aimerais pouvoir aider les humains que je rencontre à construire, designer, analyser, jouer, et communiquer de nouvelle représentations (visuel, mental et/ou conceptuel) du monde qui nous entour afin qu'on le comprenne **mieux ensemble**.

As an educator, I am committed to teaching/learning data visualization theories and practices rooted in **critical thinking** processes and **ethical design** approaches.

I want to empower students to think critically about the many factors that influence both (1) the process of designing a data visualization and (2) dataviz as a form of communication to an audience.

# I believe everyone can "do visualization".

Even though there is little formal education. Every student of visualization brings some experience of some kind and contributes a new perspective to the field of visualization. As an educator in visualization, this is what I learn from learners. It challenges me to rethink visualization each time. Our knowledge as well as approaches to it are constantly evolving as we engage with people around visualization. In my teaching, **I want students to see this richness, to be passionate about what they do, and to think critically, creatively, and independently.**

I will empower people from diverse backgrounds to **embrace the uncertainties** in data and leverage them for critical thinking through data visualization.

I will always contribute and strive for constructing an **inclusive, inviting, fair, and tolerant** VIS learning space.

I value teaching students the ethics of conducting research, including **valuing different contributions**, particularly when working in an inter/multi-disciplinary group.

I envision a strong VIS community whose **core values** are excellence, rigor, but also creativity, openness, diversity, and inclusion.

# As an educator, I am committed to reflecting along with students on the power and beauty of data visualisation, and the responsibility that comes with it.

We provide motivation by demonstrating reasons for chosen teaching goals.

We teach students their responsibility to produce expressive, efficient and effective visualizations.

We strive to offer students the chance to actively participate and contribute to their own learning experience.

We acknowledge the variety of solutions in (teaching) visualization.

We aim to recognize every student's individual learning progress by helping them achieve their individual visualization visions and learning goals.

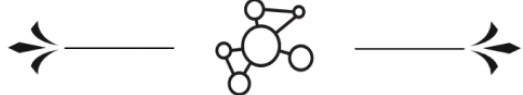

I believe...

**COMMIT**: Make personal commitments to actions or intentions that guide you to empower people through visualization.

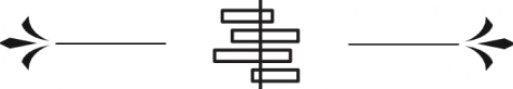

I will...

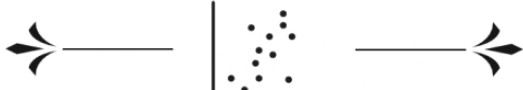

## ME-IFESTO SHARING

Now that you have made your me-ifesto, we'd like you to consider sharing it. You choose how: publicly online, privately with colleagues, in your classroom, or perhaps pinned to a bulletin board. Whatever you choose, the key is to inform and inspire others as they take the same journey you have and to begin a dialogue about how and why we empower others with data visualization.

## CONCLUSION

Well, we are not in a position to conclude anything yet. We're partway through a process and hope that you have contributed to it by doing the thinking that we asked of you in the me-ifesto exercise. Despite not being in a position to conclude, however, we do think that we have made some progress in terms of thinking about and developing our values and approaches to teaching visualization. Evidently, we think that this is worth reporting. This is our effort to get others involved in some of this thinking and the discussion around it.

So... the exercise?

## did you do it?

**If not, might you do it in the future?**
**If so, might you do it in the future again?**
We hope so — we found it very valuable.

Or maybe you are struggling to write a commitment or a value statement?

If you are uncertain about committing to a me-ifesto, might we offer you permission to stop and acknowledge where you are. Try something else to help. Perhaps imagine an acknowledgment, an invitation, an aspiration, a toast, a question, that might guide your practice as a teacher or a learner.

We might not have a single agreed manifesto that we can all commit to and sign. However, we do all feel that we know ourselves and each other, our values and how we do or do not fit in, much more fully. We were actually quite surprised: despite our various collective efforts to democratise data visualization and attendance at a workshop on empowerment, few of us had explicit statements of value and intent — or commitments to behave in particular ways — that addressed these objectives. What we have is still partial, but we have begun to discuss these issues and feel that this is taking us somewhere. Our developing chaotic collective manifesto has certainly helped us connect and think about visualization empowerment and education in new and interesting ways that feel significant and important.

We therefore prefer to end on a series of open questions on the value of this exercise, on how we might take it forward, and where it might lead.

- How can we move forward with these new reflections and possibly realizations?

- What effects might the introduction of reflective practices to visualization and analysis have in the long term?

- Do they entrench or do they inspire change? Do they evolve?

- How can we use these kinds of reflective practices to support teaching and learning in VIS?

- What does this tell us about visualization teachers and their values, motivations, and methods for empowerment?

- What might this tell us about the visualization learners who we strive to empower?

In short, this paper isn't really about us, it's about you — your values, approaches, commitments, and actions. Finding them, exposing them, questioning them, developing them. So, dear reader, valued *alt.vizzer*, you are the most important part of these questions now. Even though we don't know where this will all end up, we feel as though it's likely to take us somewhere interesting. Somewhere that helps us help the learners who we strive to empower through our visualization activity and teaching. We'd like to know how it feels to you, whether you agree, whether you have counter-arguments or different experiences and whether we can get you on board. Let us know what you come up with. Let us know what you think. Let us know whether this feels at all empowering for you. For those who you teach the *me-ifesto* to, please know that it is not an instruction, but an invitation to engage in discussion and thinking around empowerment in, for, and through visualization in teaching and learning.

## ACKNOWLEDGMENTS

The authors wish to thank the Dagstuhl Seminar 22261, "Visualization Empowerment: How to Teach and Learn Data Visualization," its organizers (Benjamin Bach, Sheelagh Carpendale, Uta Hinrichs, and Samuel Huron), and participants for their company, collegiality, ideas, deep discussions and their intense passion for teaching that so inspired us in Saarland. We also thank the anonymous reviewer who contributed a me-ifesto statement to our collection.

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
