# OpenReview forum: "Me-ifestos for Visualization Empowerment in Teaching (and Learning?)"
_IEEE.org/2022/Workshop/altVIS — Accept_

### Official Review · Reviewer_6QYY · 2022-08-05

**Review:**

The work in thought-provoking, and definitely "alternative" in its structure.

I have a few minor comments and suggestions:

- "Make personal commitments to actions or intentions that guide you to empower people through visualization" ->  "Make personal commitments to actions or state intentions that guide you to empower people through visualization"
    [You can have an intention, but I don't think it's meaningful to make a commitment to an intention]

- "right here (left, here)" - I think this is a joke that I initially missed. Consider deleting this joke, and swapping the two columns of the page, so that the reader first encounters the text telling them to do the exercise, and then encounters the instructions for what to do.

- "values and/or commitments you want to make" -> "values you hold and/or commitments you want to make"

- it would be helpful for the reader if the titles of the manifestos on page 4 were clickable links to online copies of the manifestos

- the inclusion of statements in languages other than English is a nice reflections of the diversity of the participants, but it would be helpful to also include English translations beneath these

- the "I believe..." and "I will..." boxes could be made larger

- ideally all of the references would include either a URL or a DOI link; a few are currently missing

**Conflicts:**

I am in the same research group as Benjamin Bach and Uta Hinrichs, with whom I have co-authored publications.

**Review Inclusion:**

No

**Sufficiently Alt:**

Yes

**Superlative:**

* Largest number of opinions
* Most internal disagreement

---

### Official Review · Reviewer_mCz6 · 2022-08-11

**Review:**

This paper invites the reader to reflect on their own intentions and commitments for teaching data visualizations and to create a personal manifesto (‘me-ifesto’). This work was inspired by experiences during the Dagstuhl seminar where the participants attempted to create a shared manifesto, and instead realized that each person’s individual experiences and motivations led to differences in their values and commitments, leading the group to instead create me-ifestos. The paper outlines a Me-ifesto Exercise and provides many examples of others’ me-ifestos to help the reader create their own me-ifesto.

POSITIVES:

The originality and significance of the paper is great. I agree with the authors that creating a me-ifesto is an important and inspiring activity that can guide personal teaching practices. To the best of my knowledge, there is not another paper in the area of data visualization that explains and invites readers to create a personal manifesto for their teaching practice. The paper started strong with a compelling introduction. There is a lot of literature in the VIS community focused on teaching data visualization: many books, at different levels (e.g., K-12 or novices), online or in-person courses. This activity seems like it could be helpful for instructors who teach data visualization, and to encourage a discussion across the visualization community of shared values and inspire commitments.

The design of the paper is beautifully done, with quotes arranged nicely, visual hierarchy and highlighting, and large (but not sparse) spaces dedicated to the important part --- the reader creating their own value and commitment statements.
I enjoyed the process of creating a me-ifesto. I did do the thinking and the writing and the reflecting with my coffee and I found it inspiring to read others’ me-ifestos and found it valuable to reflect on my past experiences with teaching and data visualization and my own personal values and commitments. Here is my me-ifesto (work in progress): I want to empower students to think critically about the many factors that influence both (1) the process of designing a data visualization and (2) dataviz as a form of communication to an audience. I’m looking forward to hearing others’ me-ifestos and to iterate and update my own in the future.

NEGATIVES:

The inspiration and counter-inspiration section could have done much more to provide a background on previous manifestos. I understand that it’s difficult to concisely describe key aspects of manifestos that could be large (e.g., Data Feminism is a whole book), but the paper doesn’t even try to provide any details on what these manifestos are, instead just giving us the title and author and leaving it up to the reader to do a lot of work to track down what these manifestos are.

The introduction describes what happened after these existing manifestos were collected, “we took time to read them, process them, share them and anchor some individual and collective reflection. We thought about their contents, discussed their approaches, decided what they taught us about ourselves, our own experiences and attitudes, and identified statements that resonated with us individually,” yet the paper skips all of this meaty middle part and only publishes the final me-ifestos. I was excited about the process!! And very disappointed when there was hardly anything about it in the rest of the paper. I was hoping to see (1) some descriptions of this process as it happened in the Dagstuhl Seminar, or a few descriptions of individual authors’ reflections, stories, and process of making their me-ifestos and (2) more guidance on how to reflect on these manifestos, prompts for thinking about one’s own prior experiences and how they could shape values. I was hoping the sections for CRAFT and COMMIT would have a more substantial prompt on how to craft a value/commitment statement. At the very least, I did not expect it to have a less substantial prompt than the very brief Me-ifesto Exercise description. I had to flip back several pages to remind myself what it was that I was supposed to be doing in this particular section.

I was a little disappointed that the CONSIDER step came after the CHECK step. I wanted to think about these prompts about my values, commitments, actions, intentions before I was influenced by others’. In the classroom activity called “think, pair, share,” the individual thinking comes first. It also seems contrary to the original process as described in the introduction -- “Seeing others’ statements helped us understand each other in ways that surprised us. It made us want to revise or add to some of our own.”

SUMMARY:

Although I liked the idea and design of the paper, I was disappointed with the minimalism of the inspiration section (published manifestos) and lack of guidance for the CRAFT and COMMIT activities. The process of creating a me-ifesto and reflecting on personal values and commitments is a powerful and useful one, however, this paper falls short on guiding the reader to do so effectively. I hope that the authors will revise this paper and add more to it, as I believe it can be a valuable contribution to the VIS community.

Requested revisions:

* Expand Inspiration section to give descriptions or backgrounds of what each manifesto is

* Add more stories or anecdotes for the process of creating a me-ifesto, examples of how personal experiences have influenced values, or how reading others’ me-ifestos have inspired and shapes your own

* Add more guidance or prompts for creating a value statement and a commitment statement


**Conflicts:**

N/A

**Review Inclusion:**

No

**Sufficiently Alt:**

Yes

**Superlative:**

Most homework assigned

---

### Official Review · Reviewer_B2fG · 2022-08-24

**Review:**

The research elaborates on how an in-person conversation where a group of visualization researchers share their manifestos inspires each other for future research and collaborations.
Pros:
1. the topics on empathetic conversations are innovative
2. the authors demonstrate a comprehensive guideline for global scholars to host intellectual conversations for visualization researchers.


Cons:
1. the figure qualities and demonstration need to be improved
2. the paper shall include discussions on relevant literature or events
3. the authors can shorten the paper



**Conflicts:**

NA

**Review Inclusion:**

Yes

**Sufficiently Alt:**

Yes

**Superlative:**

Most empathetic

---

### Official Review · Reviewer_T3wT · 2022-08-31

**Review:**

Alt-Meta Review:

As an alt-Meta, I am happy to say that this paper has been accepted for the 2022 alt.vis workshop! All reviewers agree that the ideas presented along with the style of presentation are alt-y and thought-provoking. I encourage you to consider how you may conduct a shorter brainstorming session at the workshop – or are there other ways to encourage people to partake in creating a me-ifesto, like picking 5 cards from a stack of values for instance?

Echoing Reviewer #2, I would also strongly encourage the authors to consider adding the following to the paper:
(1)	more description in the INSPIRATION section to provide the reader with a taste of other manifestos;
(2)	more guidance on how to create the me-ifesto and/or how to then discuss with colleagues


**Conflicts:**

I know one of the authors and also interviewed one of the authors for a study.

**Review Inclusion:**

No

**Sufficiently Alt:**

Yes

**Superlative:**

Most Likely to Walk Around with Cup of Coffee

---

### Decision · Program_Chairs · 2022-08-31

Accept